# Cellular and Humoral Immunity after the Third Vaccination against SARS-CoV-2 in Hematopoietic Stem-Cell Transplant Recipients

**DOI:** 10.3390/vaccines10060972

**Published:** 2022-06-18

**Authors:** Laura Thümmler, Michael Koldehoff, Neslinur Fisenkci, Leonie Brochhagen, Peter A. Horn, Adalbert Krawczyk, Monika Lindemann

**Affiliations:** 1Institute for Transfusion Medicine, University Hospital Essen, University of Duisburg-Essen, 45147 Essen, Germany; laura.thuemmler@uk-essen.de (L.T.); neslinurfisenkci@gmail.com (N.F.); peter.horn@uk-essen.de (P.A.H.); 2Department of Infectious Diseases, West German Centre of Infectious Diseases, University Hospital Essen, University of Duisburg-Essen, 45147 Essen, Germany; leonie.brochhagen@uk-essen.de (L.B.); adalbert.krawczyk@uk-essen.de (A.K.); 3Department of Hematology and Stem Cell Transplantation, University Hospital Essen, University of Duisburg-Essen, 45147 Essen, Germany; michael.koldehoff@uk-essen.de; 4Department of Hygiene and Environmental Medicine, University Hospital Essen, University of Duisburg-Essen, 45147 Essen, Germany

**Keywords:** ELISpot, variants of concern, vaccination response, T cells

## Abstract

Protecting vulnerable groups from severe acute respiratory syndrome coronavirus type 2 (SARS-CoV-2) infection is mandatory. Immune responses after a third vaccination against SARS-CoV-2 are insufficiently studied in patients after hematopoietic stem-cell transplantation (HSCT). We analyzed immune responses before and after a third vaccination in HSCT patients and healthy controls. Cellular immunity was assessed using interferon-gamma (IFN-γ) and interleukin-2 (IL-2) ELISpots. Furthermore, this is the first report on neutralizing antibodies against 11 variants of SARS-CoV-2, analyzed by competitive fluorescence assay. Humoral immunity was also measured by neutralization tests assessing cytopathic effects and by ELISA. Neither HSCT patients nor healthy controls displayed significantly higher SARS-CoV-2-specific IFN-γ or IL-2 responses after the third vaccination. However, after the third vaccination, cellular responses were 2.6-fold higher for IFN-γ and 3.2-fold higher for IL-2 in healthy subjects compared with HSCT patients. After the third vaccination, neutralizing antibodies were significantly higher (*p* < 0.01) in healthy controls, but not in HSCT patients. Healthy controls vs. HSCT patients had 1.5-fold higher concentrations of neutralizing antibodies against variants and 1.2-fold higher antibody concentrations against wildtype. However, half of the HSCT patients exhibited neutralizing antibodies to variants of SARS-CoV-2, which increased only slightly after a third vaccination.

## 1. Introduction

Since the beginning of the SARS-CoV-2 pandemic, over 500 million people have been infected, and more than six million people have died due to COVID-19. Stem-cell transplant recipients have increased mortality and morbidity from COVID-19 due to immunosuppression [1,2,3,4].

For individuals belonging to vulnerable groups, protection against SARS-CoV-2 infection through vaccination is of enormous importance [5]. It is equally important that people in their immediate environment are protected by vaccination, and that infection is avoided [6].

However, studies showed that immunocompromised patients developed no or only weak immune responses after SARS-CoV-2 infection and two vaccinations [2,7]. Similarly, immunocompromised individuals suffer more frequently from vaccine breakthrough infection, i.e., COVID-19 despite two vaccinations [8,9].

Studies have shown that a third vaccination is needed, particularly in vulnerable groups, to provide a protective effect against SARS-CoV-2 or to boost the existing weak immune response [10,11,12,13,14]. Currently, there are limited data on whether a third vaccination can boost specific immunity in immunocompromised patients after hematopoietic stem-cell transplantation (HSCT). Our study focuses on an even more detailed investigation of cellular and humoral immunity, particularly with regard to the evolved variants of SARS-CoV-2, which now account for the majority of infections.

We examined the cellular and humoral immunity to SARS-CoV-2 in immunocompromised and non-immunocompromised vaccinated individuals before and after the third vaccination. Cellular immunity was tested by SARS-CoV-2-specific interferon (IFN)-γ and interleukin (IL)-2 ELISpot assay and a newly established fluorescence ELISpot assay which can detect IFN-γ and IL-2 simultaneously. Titers for neutralizing antibodies to wildtype SARS-CoV-2 were analyzed by neutralization test. In addition, we examined neutralizing antibodies to variants and mutations of SARS-CoV-2 by competitive fluorescence assay. SARS-CoV-2-specific IgG antibodies were measured by semi-quantitative and quantitative ELISA.

## 2. Materials and Methods

### 2.1. Volunteers

As a first group, we included 24 HSCT patients tested prior to the third vaccination and 18 HSCT patients tested after the third vaccination against SARS-CoV-2. HSCT was performed at a median of 4.1 years (range 0.5–24) prior to blood sampling. The subgroup tested prior to the third vaccination contained 10 males and 14 females; the median age was 62 years (range 39–73). In this group, the second vaccination took place at a median of 174 days (range 55–255) prior to testing. The subgroup tested after the third vaccination contained 10 males and eight females; the median age was 61 years (range 21–71). The group was examined at a median of 34 days (range 11–95) after the third vaccination.

As a second group, we included 18 healthy volunteers before and 19 healthy volunteers after the third vaccination. The subgroup before third vaccination was composed of seven males and 11 females, and the median age was 50 years (range 31v83). They were tested at a median of 187 days (range 60–273) after the second vaccination. The subgroup after the third vaccination contained 10 males and 19 females; the median age of the donors was 50 years (range 35–65). The group was examined at a median of 42 days (range 27–72) after the third vaccination. Both groups, the HSCT patients and the healthy controls prior to and post third vaccination, were unpaired.

The study was approved by the local ethics committee of the University Hospital Essen, Germany (20-9225-BO and 20-9254-BO), and all volunteers provided informed consent to participate in the study. The study was performed in accordance with the ethical standards noted in the 1964 Declaration of Helsinki and its later amendments or comparable ethical standards.

### 2.2. CoV-iSpot for Interferon-γ and Interleukin-2

For simultaneous staining of IFN-γ and IL-2, we used the CE-marked CoV-iSpot (AID, Strassberg, Germany) in 63 samples (22 HSCT patients, 41 healthy controls). This fluorescence ELISpot (Fluorospot) includes a peptide mix of the wildtype SARS-CoV-2 spike protein. In the negative controls, we could detect, on average, 0.9 spots (range 0–4) in the group of HSCT patients and 1.1 spots (range 0–10) in healthy controls. In positive controls, we saw, on average, 383 spots (range 105–768) in the group of HSCT patients and 508 spots (range 170–925) in healthy controls. The cutoff definition was described in a previous study [15]. We chose a spot increment of seven as a cutoff for the positivity for IFN-γ and a spot increment of three for IL-2.

### 2.3. In-House ELISpot Assay

To further assess cellular immunity to SARS-CoV-2, we separately performed IFN-γ and IL-2 ELISpot assays. The procedure for IFN-γ ELISpot assay was previously described [15,16]. Briefly, 60 µL of monoclonal antibodies against IFN-γ (10 µg/mL of clone 1-D1K, Mabtech, Nacka, Sweden) or IL-2 (10 µg/mL, clone MT2A91, Mabtech) were used for coating of plates containing polyvinylidene difluoride (PVDF) membranes (MilliporeSigma™ MultiScreen™ HTS, Fisher Scientific, Schwerte, Germany) after activation with ethanol. Thereafter, ELISpot plates were washed and blocked for 30 min at 37 °C with 150 µL of AIM-V^®^ (Thermo Fisher Scientific, Grand Island, NY, USA). After incubation, AIM-V^®^ was discarded, and duplicates of 250,000 peripheral blood mononuclear cells (PBMCs) were cultured with or without of either PepTivator^®^ SARS-CoV-2 protein S1/S2 or protein S1 (600 pmol/mL of each peptide, Miltenyi Biotec, Bergisch Gladbach, Germany) in 150 µL of AIM-V^®^. In parallel, cell cultures of 250,000 PBMC were grown in the presence or absence of a protein S1 (4 µg/mL, Sino Biological, Wayne, PA, USA) in 150 µL of AIM-V^®^ for 19 h at 37 °C. After incubation, the ELISpot plates for IFN-γ were washed and incubated for 1 h with 50 µL of the alkaline phosphatase-conjugated monoclonal antibody against IFN-γ (clone 7-B6-1, Mabtech), diluted 1:200 with PBS plus 0.5% bovine serum albumin (BSA).The ELISpot plates for IL-2 were washed and incubated for 1 h with 50 µL of the biotinylated monoclonal antibody against IL-2 (clone MT2A91, Mabtech), diluted 1:100 with PBS plus 0.5% BSA. Thereafter, the plates were washed and incubated for 1 h with 100 µL of the alkaline phosphatase-conjugated streptavidin (Mabtech), diluted 1:1000 with PBS plus 0.5% BSA. After further washing of the IFN-γ and IL-2 ELISpot, 50 µL of NBT/BCIP was added, and purple spots appeared within 7 min. Spot numbers were analyzed by an ELISpot reader (AID Fluorospot, Autoimmun Diagnostika GmbH, Strassberg, Germany). Mean values of duplicate cell cultures were considered. SARS-CoV-2-specific spots were determined according to the spot increment as previously described [15]. The negative controls for IFN-γ had, on average, 0.5 spots (range 0–13) in the group of HSCT patients and 1.1 spots (range 0–10) in the group of healthy controls. The positive controls had, on average, 492 spots (range 91–600) in the group of HSCT patients and 512 spots (range 100–600) in the group of healthy controls. If the reader could not detect individual spots, we declared this as 600 spots. We chose five as cutoff for positivity for IFN-γ and seven for IL-2.

### 2.4. Assessment of Neutralizing Antibodies by Competitive Immunofluorescence

A commercial competitive immunofluorescence assay (Bio-Plex Human SARS-CoV-2 Variant Neutralization Antibody 11-Plex Panel, BIO-RAD) was used to detect neutralizing antibodies against wildtype SARS-CoV-2 and 11 mutations belonging to variants of SARS-CoV-2 (Table 1). In this commercial assay, neutralizing antibodies from serum compete with biotinylated ACE2 receptors. Detection is achieved by the addition of streptavidin–phycoerythrin (SA–PE), which binds to the biotinylated ACE2 receptor. A lower measured fluorescence indicates a higher concentration of neutralizing antibodies. Results ≥1000 ng/mL define the upper limit of the system.

### 2.5. Assessment of Neutralizating Antibodies by Cytopathic Effects

An endpoint dilution assay was used to determine the neutralization titer of sera from HSCT patients and healthy controls, as previously described [17]. Serial dilutions (1:20 to 1:2560) of the respective sera were incubated with 100 TCID_50_ of SARS-CoV-2 for 1 h at 37 °C. After the incubation, the dilutions were added to confluent Vero-E6 cells in 96-well microtiter plates and incubated for 3 days. Thereafter, cells were stained with crystal violet (Roth, Karlsruhe, Germany) solved in 20% methanol (Merck, Darmstadt, Germany) and analyzed by light microscopy for the appearance of cytopathic effects (CPEs). The neutralization titer was defined as the reciprocal of the highest serum dilution at which no CPE breakthrough was observed in any of the three test wells.

### 2.6. Antibody ELISA

For the detection of SARS-CoV-2-specific antibodies, we used a CE-marked anti-SARS-CoV-2 IgG semiquantitative ELISA (Euroimmun, Lübeck, Germany) and a quantitative ELISA (anti-SARS-CoV-2-QuantiVac-ELISA, Euroimmun, Lübeck, Germany). The ELISAs were performed according to the manufacturer’s instructions automatically at a dilution of 1:100 of the sera, using the Immunomat (Virion\Serion, Würzburg, Germany). Plates were coated with wildtype recombinant SARS-CoV-2 spike protein (S1 domain). Results for the semiquantitative ELISA were given as the ratio of patient sample to control sample. An antibody ratio of >1.1 was considered positive, of ≥0.8 to <1.1 was considered borderline, and of <0.8 was considered negative. Results for the quantitative ELISA were given as BAU/mL.

### 2.7. Statistical Analysis

We used GraphPad Prism 9.3.1 (San Diego, CA, USA) software for statistical analysis. We used the Kolmogorov–Smirnov test to check for normality. Mann–Whitney test and Spearman analysis were used to correlate numerical variables (Appendix A). Two-sided *p*-values <0.05 were considered significant. To correlate the interval between HSCT and blood collection with SARS-CoV-2-specific immune responses, we used Spearman analysis (Appendix A).

## 3. Results

### 3.1. Comparison of Cellular Immunity in HSCT Patients and Healthy Controls before and after Third Vaccination

We analyzed the cellular immune response in HSCT patients and healthy controls before and after third vaccination. In the commercial CoV iSpot, there was no significant increase after the third vaccination in HSCT patients or in healthy controls, which was observed both for IFN-γ and IL-2, when cells were stimulated with the S pool of wildtype SARS-CoV-2 (Figure 1). Prior to the third vaccination, two patients out of 12 showed a positive response for IFN-γ, while three out of 12 showed a positive response for IL-2 and two out of 12 showed a positive response for both. After vaccination, the respective numbers in the HSCT patients were five out of 12 for IFN-γ, six out of 12 for IL-2, and two out of 12 for both. The spot increment for IL 2 after third vaccination differed significantly between HSCT patients and healthy controls (*p* = 0.007) (Appendix A). In addition, the spot increments for simultaneous secretion of IFN-γ and IL-2 were significantly lower in HSCT patients than in healthy controls after the third vaccination (*p* = 0.02).

Using our *in-house* ELISpot, we observed after the third vaccination no significant increase in IFN-γ or IL-2 spots in one of the two groups after stimulation with a peptide mix of S1/S2 or S1 or with an S1 protein, which is recombinantly expressed in (human) HEK293 cells (called S1 Sino hereinafter) (Figure 2). Prior to the third vaccination, seven of the 24 patients showed a positive response to the S1/S2 peptide mix, six of the 24 showed a positive response to the S1 peptide mix, and two of the 24 showed a positive response to the S1 Sino. After vaccination, the respective numbers in the HSCT patients were five out of 18 for the S1/S2 peptide mix, six out of 18 for the S1 peptide mix, and two out of 18 for S1 Sino. For the S1/S2 peptide mix, there were no significant differences between healthy controls and HSCT patients before and after the third vaccination. For the S1 peptide mix, healthy controls displayed significantly higher IFN-γ spot increment than HSCT patients before and after third vaccination (*p* = 0.01 and *p* = 0.006). When stimulated with S1 Sino, healthy controls showed a significantly higher spot increment for IFN-γ after third vaccination than HSCT patients (*p* = 0.001) (Appendix A). For IL-2, we detected no significant differences within the groups. However, healthy controls had a higher spot increment than HSCT patients after stimulation with the S1/S2 and S1 peptide mix and with S1 Sino, both before and after third vaccination (S1/S2: *p* = 0.1 and *p* = 0.0005; S1: *p* = 0.04 and *p* = 0.01; S1 Sino: *p* = 0.02 and *p* = 0.02) (Appendix A).

Summarizing the cellular data, neither HSCT patients nor healthy controls displayed significantly higher SARS-CoV-2-specific responses after the third vaccination, which was detected by various assay formats and for the cytokines IFN-γ and IL-2. However, as expected, responses in healthy controls were overall higher than in HSCT patients.

### 3.2. Comparison of Humoral Immune Responses to Variants of SARS-CoV-2 in HSCT Patients and Healthy Controls

We examined using a competitive immunoassay whether vaccination also leads to a humoral immune response against different variants and mutations of SARS-CoV-2. In HSCT patients, there was no significant increase in neutralizing antibodies after third vaccination for any of the mutations tested, whereas, in healthy controls, there was a significant increase in neutralizing antibodies for each variant/mutation tested, i.e., alpha, beta, gamma, delta (plus), epsilon, eta, iota, kappa, lambda, mu, and omicron (Figure 3). In detail, prior to the third vaccination, 15 of the 24 patients showed a positive response to the D614G mutation, found in the delta and omicron variant (Figure 3i), 15 of the 24 showed a positive response to the K417N mutation (omicron variant, Figure 3j), and 13 of the 24 showed a positive response to the N501Y mutation (omicron, Figure 3k). After vaccination, the respective numbers in the HSCT patients were 11 out of 18 for D614G, 12 out of 18 for K417N, and 11 out of 18 for N501Y. Thus, about half of the patients responded to mutations found in the delta (plus) and omicron variant. Prior to the third vaccination, eight of the nine healthy controls responded to the D614G mutation, seven of nine responded to K417N, and seven of nine responded to N501Y. After vaccination, the numbers in healthy controls were 19 out of 19 for D614G, 19 out of 19 for K417N, and 19 out of 19 for N501Y. The comparison between HSCT patients and the healthy control group after third vaccination revealed significant differences in neutralizing antibodies against all variants/mutations tested (Appendix A).

### 3.3. Comparison of Humoral Vaccination Responses to Wildtype Virus in Patients after Hematopoietic Stem-Cell Transplantation and Healthy Controls

We evaluated neutralizing antibodies against wildtype SARS-CoV-2 in the sera to determine if the immunocompromised individuals might build up a similar level of neutralizing antibodies to healthy controls. In HSCT patients, no significant increase in neutralizing antibodies after the third vaccination could be detected (mean titer^−1^ of 432 vs. 686, *p* = 0.4), whereas there was a significant increase in neutralizing antibodies in healthy controls (mean titer^−1^ of 140 vs. 884, *p* < 0.0001) (Figure 4). A significantly lower titer was observed in HSCT patients than in healthy controls after the third vaccination (*p* = 0.05).

In addition, we measured the antibody ratio and the antibody concentration in 24 HSCT patients and 18 healthy controls before the third vaccination against SARS-CoV-2, as well as in 18 HSCT patients and 19 healthy controls after the third vaccination. There was no significant difference in antibody ratios before and after the third vaccination in HSCT patients (mean ratio of 5.4 vs. 6.5, *p* = 0.6, geometric mean concentration of 195 vs. 350 BAU/mL, *p* = 0.4); however, there was in healthy controls (mean ratio of 4.8 vs. 10.6, *p* < 0.0001, geometric mean concentration of 353 vs. 2065 BAU/mL, *p* < 0.0001) (Figure 5). The antibody ratio showed a significant difference between healthy controls and HSCT patients after the third vaccination (*p* = 0.002, Appendix A).

### 3.4. Correlation of SARS-CoV-2-Specific Immune Responses with Age and Interval between HSCT and Blood Collection

We analyzed the correlation of the results of cellular and humoral immunity with age of HSCT patients and healthy controls by Spearman analyses. For the cellular immunity, we observed in the CoV iSpot significant correlations between the age and the spot increment only for IL-2, with simultaneous staining of IFN-γ and IL-2 in HSCT patients after the third vaccination (IL-2: *r* = −0.7, *p* = 0.008, IFN-γ and IL-2: *r* = −0.6, *p* = 0.03). In our *in-house* ELISpot, we detected a significant correlation between age and spot increment for IFN-γ in HSCT patients after the third vaccination after stimulation with S1 Sino (*r* = −0.5, *p* = 0.03). After stimulation with S1/S2 peptide mix and S1 peptide mix, we observed a significant correlation between age and spot increment for IFN-γ in healthy controls after the third vaccination (S1/S2: *r* = −0.8, *p* < 0.0001, S1: *r* = −0.6, *p* = 0.01) (Appendix A). For humoral immunity, we could not detect a significant correlation with age for titers of neutralizing antibodies or for antibody ratio or concentration (Appendix A).

We also performed Spearman analyses to correlate the interval between HSCT and blood collection with cellular or humoral immunity. For cellular immunity, we detected a significant correlation only for IFN-γ after stimulation with S1 Sino after the third vaccination (*r* = −0.5, *p* = 0.03) (Appendix A). For humoral immunity, we observed no significant correlation (Appendix A).

## 4. Discussion

We could not observe significant increases in cellular immunity after the third vaccination in both groups. However, healthy controls showed higher mean values for IFN-γ and IL-2 spot increments before and after the third vaccination than HSCT patients. A limitation of our study is the low frequency of SARS-CoV-2-specific cells in HSCT patients. Nevertheless, as indicated by the negative and positive controls, all assays included into our study were valid.

Our study indicates that HSCT patients and healthy controls both displayed neutralizing antibodies against variants and mutations of SARS-CoV-2 before the third vaccination against SARS-CoV-2. However, these antibodies increased only slightly after vaccination in HSCT patients, whereas healthy controls showed a significant increase. When looking at the individual values, an increase in humoral immunity was observed in about 15% of the HSCT patients. However, it should be noted that a protective effect can hardly be assumed due to the detection of specific antibodies. Dhakal et al. previously reported that only one-third of HSCT patients generated a humoral immune response against SARS-CoV-2 after second vaccination [18]. One reason for this finding is immunosuppressive therapy after stem-cell transplantation [19]. Other hematological malignancies, such as chronic lymphocytic leukemia (CLL), also led to impaired vaccination responses to SARS-CoV-2 [20]. Similarly, the timing of vaccination, immune status prior to HSCT, and type of HSCT may also play a role, as shown for other vaccines [21,22].

The results of the antibody ELISA and neutralization assay indicate that, after the third vaccination, antibody concentrations and the distribution of titers remained similar in immunosuppressed individuals, whereas a significant increase was seen in healthy controls. Other studies have also previously shown that the humoral immune response to SARS-CoV-2 vaccination is reduced in 30–60% of HSCT patients [13,14,23]. Currently, there are very few studies on immune responses after third vaccination in HSCT patients. Most studies focus on the humoral immune response, but point to the important role of cellular immunity, which we investigated [24,25,26]. Einarsdottir et al. were the only investigators of the cellular immune response. They examined 37 HSCT patients for humoral and cellular immune response 4 weeks after the third vaccination against SARS-CoV-2 by ELISA. T cells were stimulated with the N-terminus of the spike protein of SARS-CoV-2 and were measured after 48 h of incubation. Einarsdottir et al. demonstrated that 49% of tested volunteers lacked a cellular immune response after the third vaccination against SARS-CoV-2 [27]. However, they used a different method from the current study and measured the released IFN-γ in plasma following stimulation of whole-blood samples, but not PBMCs at a defined cell number. Furthermore, they did not compare responses after the second and third vacciantion in that study. In another study, they reported on T-cell responses in 50 HSCT recipients after the second vaccine dose and unexpectedly showed that 28% failed to achieve detectable T-cell responses [23]. Presumably, there were slight differences in the experiemntal settings of their two studies, because the data would otherwise imply that the fraction of patients with positive T-cell responses even declined after the third vaccination.

In a previous study on immunity toward wildtype SARS-CoV-2 [28], we investigated the humoral and cellular immune responses of HSCT patients and healthy controls before the first vaccination, after the first vaccination, after the second vaccination, and after infection. After the second vaccination, the humoral immune responses in HSCT patients increased by about a factor of 15 compared to the immune response after the first vaccination. In the current study, the humoral responses increased by a maximum factor of 1.1 in HSCT patients. In terms of cellular immune responses, the comparison showed stronger responses before the third vaccination than could be detected in the previous study at a median of 30 days after the second vaccination. It could, thus, be assumed that there may be a delayed cellular vaccination response. However, after the third vaccination, the increase in spot increment was very small. In the present study, in contrast to the previous one, no correlation was found between the strength of immune responses and the interval between HSCT and blood collection. However, because of a median interval between HSCT and blood collection of 4.1 years, a correlation was not expected. There was a negative correlation between age and immune responses. This was to be expected due to the median age of the tested groups, as immune responses decrease with age.

In conclusion, about half of the HSCT patients exhibited low levels of neutralizing antibodies to variants of SARS-CoV-2, which could not be increased significantly by a third vaccination. Moreover, cellular immunity against SARS-CoV-2, as observed in about 20% of the patients, did not increase after the third vaccination in HSCT patients. Of note, previous data indicate that vaccinated individuals retain T-cell immunity to the SARS-CoV-2 omicron variant and, thus, showed a minimal escape at the T-cell level [29]. Nevertheless, the HSCT patients should be partly protected against SARS-CoV-2, either by neutralizing antibodies to variants of SARS-CoV-2 or by cross-reactive T cells. A third dose of the vaccine should be given to HSCT patients, especially because we observed by trend an increase in neutralizing antibodies against variants of SARS-CoV-2.

## Figures and Tables

**Figure 1 vaccines-10-00972-f001:**
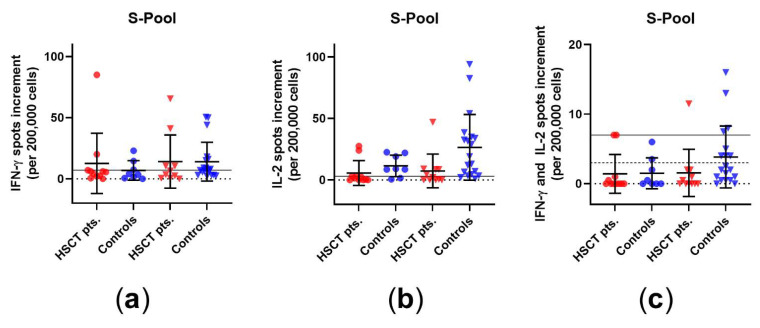
SARS-CoV-2-specific CoV-iSpot responses in HSCT patients and healthy controls before and after the third vaccination. Distribution of (**a**) IFN-γ, (**b**) IL-2, and (**c**) simultaneous IFN-γ and IL-2 CoV-iSpot responses upon stimulation with S pool of wildtype SARS-CoV-2. Please note that the scales differ. Circles indicate data before the third vaccination (red: HSCT, blue: healthy controls), while triangles indicate data after the third vaccination. Responses before and after the third vaccination were compared by two-tailed Mann–Whitney test. Horizontal lines indicate mean values, while error bars indicate the standard deviation. The dotted line represents the zero line. The black horizontal line indicates the cutoff. In panel (**c**), the continuous line indicates the cutoff for IFN-γ, while the dashed line indicates the cutoff for IL-2.

**Figure 2 vaccines-10-00972-f002:**
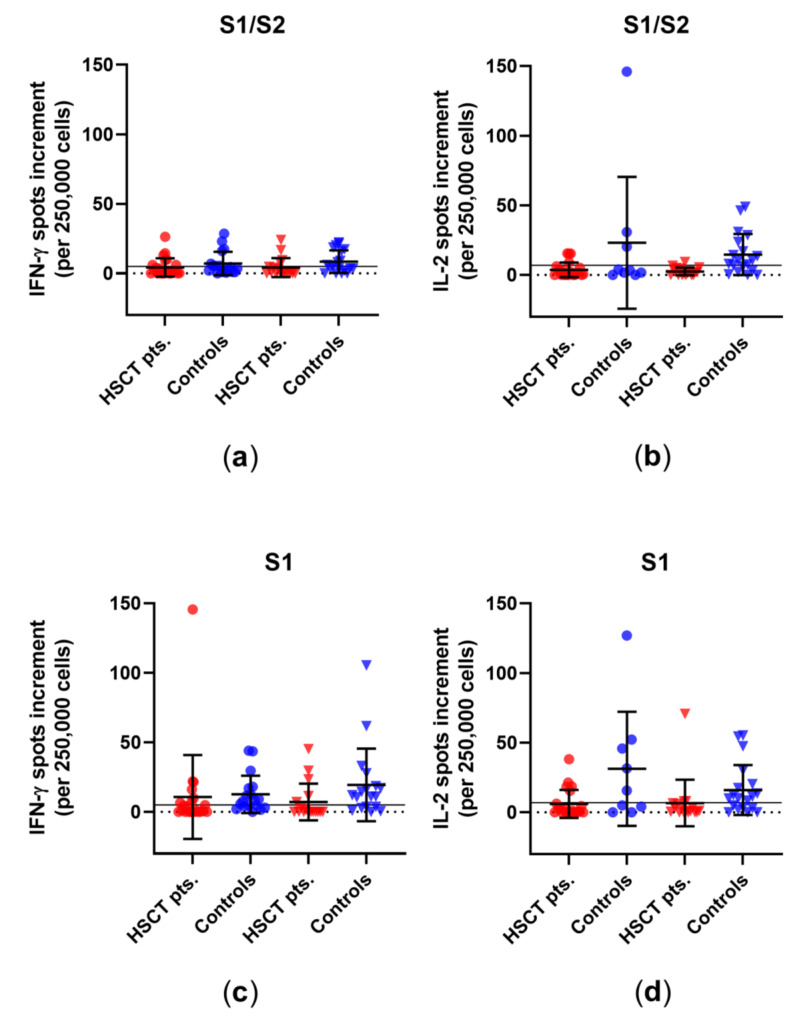
SARS-CoV-2-specific ELISpot responses in HSCT patients and healthy controls before and after third vaccination. Distribution of (**a**) IFN-γ and (**b**) IL-2 ELISpot responses upon stimulation with an S1/S2 peptide mix. Distribution of (**c**) IFN-γ and (**d**) IL-2 ELISpot responses upon stimulation with an S1 peptide mix. Distribution of (**e**) IFN-γ and (**f**) IL-2 ELISpot responses upon stimulation with S1 Sino. Please note that the scales differ. Circles indicate data before the third vaccination (red—HSCT, blue—healthy controls), while triangles indicate data after the third vaccination. Responses were compared by two-tailed Mann–Whitney test. Horizontal lines indicate mean values, while error bars indicate the standard deviation. The dotted line represents the zero line. The black horizontal line indicates the cutoff.

**Figure 3 vaccines-10-00972-f003:**
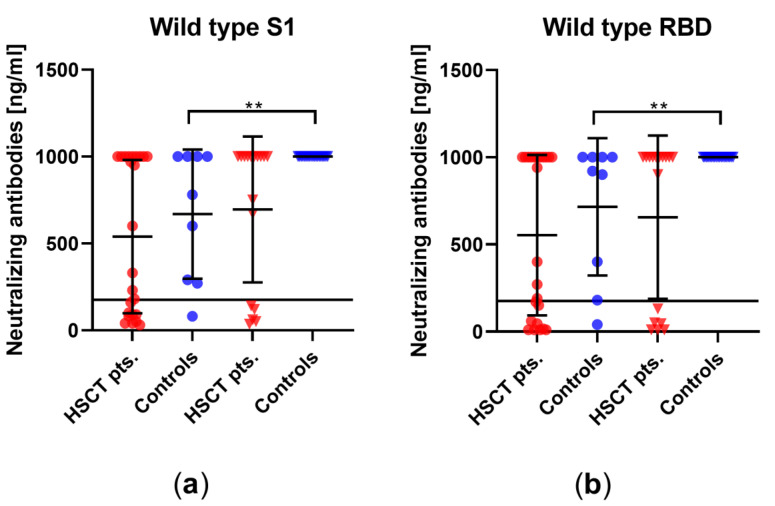
Concentration of neutralizing antibodies against different variants of the subunit 1 of spike protein (S1) or the receptor-binding domain (RBD) of SARS-CoV-2 in HSCT patients and healthy controls before and after third vaccination. Humoral responses before and after third vaccination against (**a**) wildtype S1, (**b**) wildtype RBD, (**c**) alpha, (**d**) beta, (**e**) gamma, (**f**) E484K, (**g**) epsilon, (**h**) kappa, (**i**) D614G, (**j**) K417N, and (**k**) N501Y. D614G is a mutation found in delta and omicron variants, while K417N and N501Y are mutations in the omicron variant. Circles indicate data before the third vaccination (red—HSCT, blue—healthy controls), while triangles indicate data after the third vaccination. Responses were compared by two-tailed Mann–Whitney test (** *p* < 0.01, *** *p* < 0.001, **** *p* <0.0001). Horizontal lines indicate mean values, while error bars indicate the standard deviation.

**Figure 4 vaccines-10-00972-f004:**
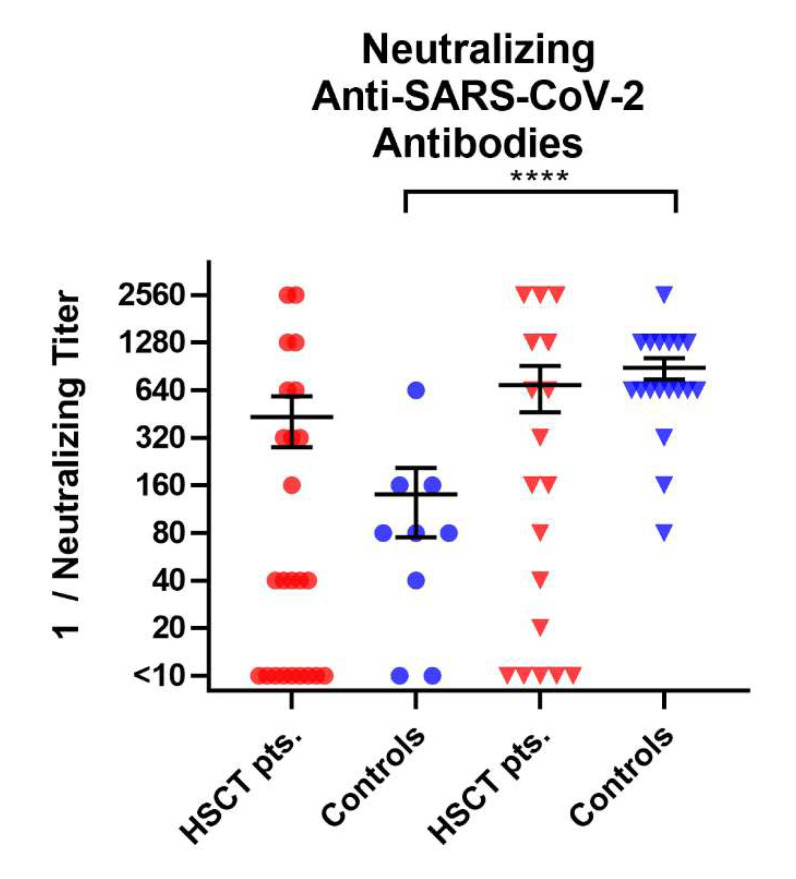
Titer of SARS-CoV-2-specific neutralizing antibodies in HSCT patients and healthy controls before and after third vaccination. The *y*-axis shows the reciprocal of the titer of neutralizing anti-SARS-CoV-2 antibodies. Circles indicate data before the third vaccination (red—HSCT, blue—healthy controls), while triangles indicate data after the third vaccination. Responses were compared by two-tailed Mann–Whitney test (**** *p* < 0.0001). Horizontal lines indicate mean values, while error bars indicate the standard deviation.

**Figure 5 vaccines-10-00972-f005:**
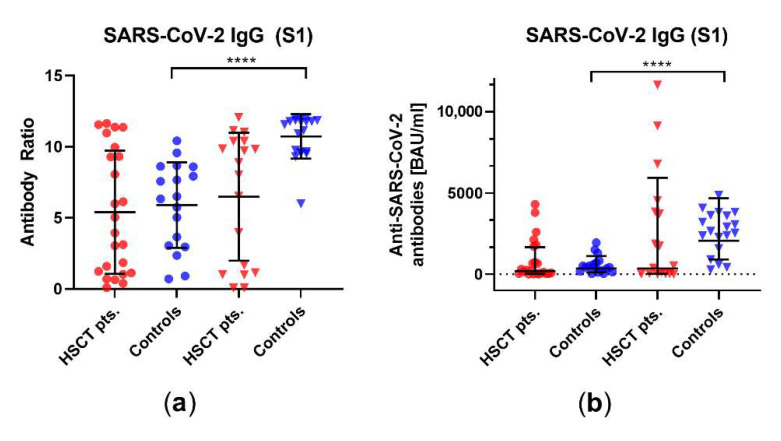
SARS-CoV-2-specific IgG antibody responses in HSCT patients and healthy controls before and after the third vaccination. SARS-CoV-2-specific IgG antibody responses are shown (**a**) as antibody ratios, which determines a quotient of antibodies in the patient samples and in a control sample, and (**b**) as SARS-CoV-2-specific IgG antibody concentrations in units of BAU/mL. Circles indicate data before the third vaccination (red—HSCT, blue—healthy controls), while triangles indicate data after the third vaccination. Responses were compared by two-tailed Mann–Whitney test (**** *p* < 0.0001). The dotted line represents the zero line. For the antibody ratio, horizontal lines indicate mean values, while error bars indicate the standard deviation. For the antibody concentration, the geometric mean and geometric standard deviation factor are indicated.

**Table 1 vaccines-10-00972-t001:** Overview of the mutations tested by the competitive immunofluorescence assay with assignment to the different variants of SARS-CoV-2.

	Mutation	HV69-70 del	Y144 del	K417N	K417T	L452R	E484K	E484Q	N501Y	A570D	D614G	P681H
Variant												
Wildtype												
B 1.1.7 (alpha)		✓	✓						✓	✓	✓	✓
B 1.351 (beta)				✓			✓		✓		✓	
P.1 (gamma)				✓			✓		✓		✓	
B 1.617.2/AY.1/.2 (delta/delta plus)						✓					✓	
B 1.427/B 1.429 (epsilon)						✓					✓	
B 1.525 (eta)		✓					✓				✓	
B 1.526 (iota)							✓				✓	
B 1.617.1/B 1.17.3 (kappa)						✓		✓				
C.37 (lambda)											✓	
B 1.621 (mu)							✓		✓		✓	✓
B 1.1.529 (omicron)		✓		✓					✓		✓	✓

## Data Availability

The data presented in this study are available on request from the corresponding author. The data are not publicly available due to privacy restrictions.

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
