# Peer review of "Cellular and Humoral Immunity after the Third Vaccination against SARS-CoV-2 in Hematopoietic Stem-Cell Transplant Recipients"

_vaccines, 2022, doi:10.3390/vaccines10060972_

Round 1

Reviewer 1 Report

This work evaluates cellular and antibody responses before and after third booster of COVID-19 vaccine in a vulnerable population that received hematopoietic stem cell transplantation. Though interesting and important, this manuscript needs extensive edits:

1. Include a flow chart showing the number of patients on either groups. I assume the individuals in healthy controls are same before and after the third vaccination. It is not clear why there are more participants after than before in healthy controls.

2. Remove all tables that shows the p values. Remove 'ns' reflecting non-significant from all figures. Instead, when there is significant differences then show the p value or asterisk between the groups. For all figures, replace the term 'dots' with 'circles'. 

3. Why neutralizing antibody data is shown in ng/mL instead of BAU/mL in figure 3? Express in international units. 

4. Not clear what is presented in figure 4. The title says 'SARS-CoV-2 IgG' while Y axis says '1/Neutralizing titer'. 

5. Figure 5 is also confusing and not clear what they are showing. Not clear which antibodies were compared while determining 'antibody ratio'. Not sure what is presented in figure 5b? 

6. In subheading 3.5, it is wrong to state correlation with 'clinical parameters' as they only compared correlation with 'age'.

Author Response

Dear Reviewer,

Please find the point by point response below. Thank you for reviewing our manuscript, your comments have improved our manuscript.

  1. Include a flow chart showing the number of patients on either groups. I assume the individuals in healthy controls are same before and after the third vaccination. It is not clear why there are more participants after than before in healthy controls.

Unfortunately, we could not test the same individuals prior to and after the third vaccination. At the time of analysis, many of the healthy controls had already been vaccinated three times, resulting in the higher number of third-time vaccinated individuals. We added the important information, that this is not a paired analysis, on page 2 line 82 (Both groups, the HSCT patients and the healthy controls prior to and post third vaccination, were unpaired.).

  1. Remove all tables that shows the p values. Remove 'ns' reflecting non-significant from all figures. Instead, when there is significant differences then show the p value or asterisk between the groups. For all figures, replace the term 'dots' with 'circles'.

Thank you for pointing this out. We have changed the tables accordingly and added the p-values to the supplement.

  1. Why neutralizing antibody data is shown in ng/mL instead of BAU/mL in figure 3? Express in international units.

The manufacturer of the assay specifies this unit. Since this is a conversion based on a standard curve, we have to leave the unit as specified by the manufacturer.

  1. Not clear what is presented in figure 4. The title says 'SARS-CoV-2 IgG' while Y axis says '1/Neutralizing titer'.

We have changed the corresponding heading (p.12: Neutralizing Anti-SARS-CoV-2 Antibodies). Furthermore, we now explain what is shown on the y-axis (page 12 line 265-266: The y-axis shows the reciprocal of the titer of neutralizing anti-SARS-CoV-2 antibodies.).

  1. Figure 5 is also confusing and not clear what they are showing. Not clear which antibodies were compared while determining 'antibody ratio'. Not sure what is presented in figure 5b?

Figure 5a shows the result of the semi-quantitative ELISA, which determines a ratio of the patient sample with a control sample. This explanation is now added to the figure legend (page 13 line 281-283: SARS-CoV-2 specific IgG antibody responses are shown (a) as antibody ratios, which determines a quotient of antibodies in the patient samples and in a control sample and (b) as SARS-CoV-2 specific IgG antibody concentrations in the unit BAU/ml.). Panel (b) shows quantitative results in the unit BAU/ml.

  1. In subheading 3.5, it is wrong to state correlation with 'clinical parameters' as they only compared correlation with 'age'.

Thank you for the comment. We have changed the headline (p. 13 line 286-287: 3.5. Correlation of SARS-CoV-2 specific immune responses with age and interval between HSCT and blood collection).

Reviewer 2 Report

The Article  Titled "Cellular and humoral immunity after the third vaccination against SARS-CoV-2 in hematopoietic stem cell transplant recipients", authored by Laura Thümmler and coleagues, provides evidence on HSC transplant recipients. The manuscript is well designed and provides interesting findings that are relevant to the SARS-COV2 pandemic. I would like to congratulate the authors for their work. My Comments and Suggestions for Authors are the following:   1) There are well presented data. However the authors go into great detail by presenting the results of Mann-Whitney test in tables 2-6. I suggest that these results should go into supplementary data.   2) Dir the authors check their data for normality? In that way thay would signify the use of Mann-Whitney test.

Author Response

Dear Reviewer,

Please find our point by point response below. Your comments on our manuscript have improved it.

The Article Titled "Cellular and humoral immunity after the third vaccination against SARS-CoV-2 in hematopoietic stem cell transplant recipients", authored by Laura Thümmler and coleagues, provides evidence on HSC transplant recipients. The manuscript is well designed and provides interesting findings that are relevant to the SARS-COV2 pandemic. I would like to congratulate the authors for their work. My Comments and Suggestions for Authors are the following:

1) There are well presented data. However the authors go into great detail by presenting the results of Mann-Whitney test in tables 2-6. I suggest that these results should go into supplementary data.

We have taken the tables from the manuscript and included them in the supplement.

2) Dir the authors check their data for normality? In that way thay would signify the use of Mann-Whitney test.

We checked the data for normality using Kolmogorov-Smirnov test. We have added this in 2.7 (p.5 line 162-163: We used Kolmogorov-Smirnov-test to check for normality. Mann-Whitney test and Spearman analysis were used to correlate numerical variables.).

Round 2

Reviewer 1 Report

Comments raised earlier are addressed.